# Combined Impact of Inflammation and Pharmacogenomic Variants on Voriconazole Trough Concentrations: A Meta-Analysis of Individual Data

**DOI:** 10.3390/jcm10102089

**Published:** 2021-05-13

**Authors:** Léa Bolcato, Charles Khouri, Anette Veringa, Jan Willem C. Alffenaar, Takahiro Yamada, Takafumi Naito, Fabien Lamoureux, Xavier Fonrose, Françoise Stanke-Labesque, Elodie Gautier-Veyret

**Affiliations:** 1Laboratory of Pharmacology, Pharmacogenetics and Toxicology, Grenoble Alpes University Hospital, 38000 Grenoble, France; Lbolacto@chu-grenoble.fr (L.B.); XFonrose@chu-grenoble.fr (X.F.); FStanke@chu-grenoble.fr (F.S.-L.); 2Pharmacovigilance Unit, Grenoble Alpes University Hospital, 38000 Grenoble, France; ckhouri@chu-grenoble.fr; 3Department of Clinical Pharmacy and Pharmacology, University Medical Center Groningen, University of Groningen, 9700 RB Groningen, The Netherlands; a.veringa@umcg.nl (A.V.); j.w.c.alffenaar@umcg.nl (J.W.C.A.); 4Faculty of Medicine and Health, School of Pharmacy, University of Sydney, Sydney, NSW 2006, Australia; 5Westmead Hospital, Sydney, NSW 2145, Australia; 6Marie Bashir Institute for Infectious Diseases, University of Sydney, Sydney, NSW 2006, Australia; 7Department of Hospital Pharmacy, Hamamatsu University School of Medicine, 1-20-1 Handayama, Higashi-ku, Hamamatsu 431-3192, Japan; yamadat@hama-med.ac.jp (T.Y.); naitou@hama-med.ac.jp (T.N.); 8Laboratory of Pharmacology, Toxicology and Pharmacogenomics, Rouen University Hospital, 76000 Rouen, France; Fabien.Lamoureux@chu-rouen.fr; 9Grenoble Alpes University, INSERM, CHU Grenoble Alpes, HP2, 38000 Grenoble, France

**Keywords:** voriconazole, therapeutic drug monitoring, inflammation, pharmacogenomics, personalized treatment

## Abstract

Few studies have simultaneously investigated the impact of inflammation and genetic polymorphisms of cytochromes P450 2C19 and 3A4 on voriconazole trough concentrations. We aimed to define the respective impact of inflammation and genetic polymorphisms on voriconazole exposure by performing individual data meta-analyses. A systematic literature review was conducted using PubMed to identify studies focusing on voriconazole therapeutic drug monitoring with data of both inflammation (assessed by C-reactive protein level) and the pharmacogenomics of cytochromes P450. Individual patient data were collected and analyzed in a mixed-effect model. In total, 203 patients and 754 voriconazole trough concentrations from six studies were included. Voriconazole trough concentrations were independently influenced by age, dose, C-reactive protein level, and both cytochrome P450 2C19 and 3A4 genotype, considered individually or through a combined genetic score. An increase in the C-reactive protein of 10, 50, or 100 mg/L was associated with an increased voriconazole trough concentration of 6, 35, or 82%, respectively. The inhibitory effect of inflammation appeared to be less important for patients with loss-of-function polymorphisms for cytochrome P450 2C19. Voriconazole exposure is influenced by age, inflammatory status, and the genotypes of both cytochromes P450 2C19 and 3A4, suggesting that all these determinants need to be considered in approaches of personalization of voriconazole treatment.

## 1. Introduction

Voriconazole (VRC) is a broad-spectrum azole antifungal agent indicated for the treatment and prevention of invasive fungal infections. It is one of the first-line treatments for invasive aspergillosis [1,2]. However, despite adequate care, mortality due to invasive aspergillosis remains very high, ranging between 19 and 61% for patients with hematological malignancies [3]. One of the possible causes of these many failures is insufficient exposure to the drug [4]. Indeed, VRC exhibits high pharmacokinetic variability, with insufficient concentrations exposing patients to an increased risk of treatment failure and excessively high concentrations resulting in adverse effects and the risk of treatment discontinuation [5,6]. In this context, VRC therapeutic drug monitoring (TDM) is recommended throughout treatment [2,7].

As invasive aspergillosis is a serious and directly life-threatening disease, it is essential to achieve effective concentrations as soon as treatment is initiated [8,9,10]. In this context, VRC personalized treatment with a priori dose adjustment has already been proposed instead of five days after the initiation of treatment [4]. Such strategies are most frequently based on the cytochrome P450 (CYP) 2C19 genotype [11,12,13,14], as many studies have demonstrated the contribution of CYP2C19 polymorphisms in the variability of VRC trough concentrations (Cmin) [15,16,17,18].

In addition to CYP2C19 polymorphisms, numerous other factors are known to influence VRC exposure. For example, certain genetic variants of CYP3A4 (rs35599367 and rs1464637) [19,20,21] and inflammatory status [22,23,24,25,26] are associated with increased VRC Cmin. The clinical implications have been limited, as only a few small-scale studies have simultaneously evaluated genetic polymorphisms of both CYP2C19 and CYP3A4, along with the inflammatory status of patients [25,26,27,28]. Moreover, it was suggested that the effect of inflammation on VRC exposure may depend on the CYP genotype [27,28].

We aimed, therefore, to more precisely define the respective impact of the inflammatory state and genetic variants on VRC exposure by gathering available data to perform an individual patient data meta-analysis.

## 2. Materials and Methods

This meta-analysis was performed according to the Preferred Reporting Items for Systematic Reviews and Meta-analysis statement guideline [29]. The study is registered with PROSPERO (CRD42020162292), and the protocol and systematic search strategy are available online.

### 2.1. Systematic Literature Review

A systematic literature review was conducted using Pubmed to identify studies evaluating the impact of CYP genetic polymorphisms on VRC exposure, taking into account the inflammatory status of patients. The following search terms were used: (“VORICONAZOLE”) AND (“PHARMACOGENETICS” OR “PHARMACOGENOMICS” OR “THERAPEUTIC DRUG MONITORING” OR “CYP2C19” OR “CYP3A4” OR “PHARMACOKINETICS” OR “POLYMORPHISM”). The terms (“VORICONAZOLE”) AND (“THERAPEUTIC DRUG MONITORING” OR “PHARMACOKINETICS”) AND (“INFLAMMATION”) were used to identify studies that evaluated the impact of inflammation on VRC concentrations.

### 2.2. Study Selection Criteria

One of the authors (LB) first screened studies based on titles and abstracts. Then, a second selection was made by two authors (LB and EG) based on the full text of the manuscripts. Studies were deemed eligible if they were case-control or cohort studies with VRC Cmin measured at pharmacokinetic steady-state and genetic data (at least CYP2C19 genotyping ± CYP3A4 and 3A5 genotyping) were available. Even if the inflammatory status of patients was not investigated in some studies, it was not a discriminating criterion for inclusion because CRP levels determined during routine medical care could be retrospectively collected. Literature reviews, case reports, and studies conducted on pediatric populations, healthy volunteers, or less than 10 patients were excluded.

### 2.3. Data Extraction

The objective of this work was to collect individual patient data from the studies selected during the systematic literature review. Thus, all the authors were contacted three times by email. The following data were requested and merged into a single database for analysis: patient identification, age, sex, weight, main pathology, CRP level, method of measuring CRP, liver enzymes (aspartate amino transferase (ASAT) and alanine aminotransferase (ALAT)), VRC Cmin determined at pharmacokinetic steady-state, the method for VRC Cmin determination, date of blood collection, date of VRC initiation, daily dose, route of administration, concomitant proton-pump inhibitor (PPI) treatment, CYP2C19, 3A4, and 3A5 genotype, and genotyping technique. The inclusion criteria were patients with a CYP2C19 genotype ± CYP3A4 and CYP3A5 genotypes with at least one pair of VRC Cmin and CRP level determined concomitantly. The absence of major drug-drug interactions was verified in each included study. The presumed ethnicity of the patients was determined based on the geographical origin of the studies, except for one study in which ethnicity was specified [30]. VRC Cmin values below the limit of quantification were replaced by the lower limit of quantification of the method.

The phenotype of CYP2C19 (poor (PM), intermediate (IM), extensive (EM), rapid (RM), or ultrarapid (UM) metabolizer) was determined based on CPIC recommendations for each patient [31]. Subsequently, three groups were defined: patients with increased metabolic capacity (RM and UM), patients with decreased metabolic capacity (IM and PM), and patients with standard metabolic capacity (EM). Similarly, the phenotype of CYP3A4 and 3A5 was determined for each patient if data were available. For CYP3A4, patients with the rs35599367 (CYP3A4*22) allele were assigned an IM phenotype. For CYP3A5, the rs776746 (CYP3A5*1) allele is associated with the expression of this cytochrome, unlike the *3 alleles, which is associated with non-expression. The combined genetic score, including the CYP2C19, CYP3A4, and CYP3A5 genotypes, was calculated, as previously described (see [19] and Appendix A).

### 2.4. Quality Assessment

The quality of the included studies was assessed by two authors (LB and CK) using the Newcastle-Ottawa Scale (NOS), as they were observational case-control or cohort studies. This scale assigns a maximum of 9 stars for good quality studies with a low risk of bias. It is based on the selection of study groups, comparability of groups, and determination of the exposure or outcome of interest for case-control or cohort studies.

### 2.5. Statistical Analysis

A linear mixed-effect model was used to assess the influence of various factors on VRC exposure. In the base mixed-effect model, random effects were included in the intercept for interindividual variability and study. Then, we performed univariate analyses for all continuous (age, weight, ASAT, ALAT, CRP levels, daily dose, and combined genetic score) and categorical (route of administration, concomitant PPI intake, CYP3A4, 3A5, and 2C19 genotypes) variables. Covariates associated with a *p*-value < 0.1 in the univariate analysis were considered clinically relevant and biologically plausible and were, therefore, included in the multivariate intermediate model. The final model was selected using a backward stepwise process based on the Akaike information criterion. Finally, all assumptions were checked in the final model, including linearity, absence of collinearity, homoscedasticity, normality of residuals, absence of influential data points, and independence. Finally, the marginal means of the selected variables in the final model were plotted. Missing data for weight, ASAT, and ALAT were imputed using subject-centered means of available data from other visits.

The genetic score was included in the final model but was not computable for all studies due to the lack of data for the CYP3A genotype. We, therefore, performed two sensitivity analyses by substituting the genetic score with the CYP2C19 phenotype alone and the CYP2C19 and CYP3A4 phenotypes. A *p*-value < 0.05 was considered statistically significant. All statistical analyses were performed using Jamovi (version 1.1.9) and R (version 3.6.1).

## 3. Results

### 3.1. General Characteristics of Studies

The study selection process is shown in Figure 1. Among the 1793 articles initially identified, 85 were selected for full-text evaluation. Forty-six articles were selected after the exclusion of 39 that did not meet the inclusion criteria. Eleven of the 46 corresponding authors who were approached responded to our emails, resulting in the collection of individual data from six studies.

The main characteristics of the studies included in the meta-analysis are summarized in Table 1. Three studies were on retrospective cohorts [15,19,30], two were prospective observational studies [26,28], and one a retrospective case-control study [25]. The total number of included patients was 203 for 754 VRC Cmin values. The genotypes of both CYP2C19 and CYP3A4/5 were determined for 4/6 studies, corresponding to 136/203 (67.0%) patients. All characteristics of the patients, including the frequency of various phenotypes for each CYP, are summarized in Appendix A. The combined genetic scores of various CYP2C19 and CYP3A4/5 polymorphisms are presented in Appendix A.

### 3.2. Determinants of Voriconazole Trough Concentration

The results of the univariate analysis are presented in Table 2. Weight, liver function, concomitant treatment by PPI, and CYP3A5 phenotype were not associated with the VRC Cmin. Conversely, age, VRC daily dose, and CRP levels were significantly associated with a higher VRC Cmin, whereas oral VRC administration was significantly associated with a lower VRC Cmin. The phenotypes of CYP2C19 (RM/UM versus EM) and CYP3A4 were associated with VRC Cmin, even if statistical significance was not reached for CYP3A4. Similarly, the combined genetic score was associated with VRC Cmin, with a lower VRC Cmin for a higher combined genetic score.

Results of the multivariate analysis are presented in Table 3, which shows the results of three different linear mixed-effect models. All three models showed that higher age, VRC daily dose, and CRP levels were significantly and independently associated with a higher VRC Cmin. Model 1, based on the largest number of observations, showed the CYP2C19 phenotype to be significantly associated with variations of VRC Cmin. Model 2, which individually considered each phenotype of CYP2C19 and 3A4, showed that the phenotypes of both CYPs significantly influence VRC Cmin. Model 3, which considered the combined genetic score, showed that an increase in the genetic score is significantly associated with a decrease in the VRC Cmin. An increase of 1 unit of the combined genetic score was associated with a reduction of 43% of the VRC Cmin, whereas an increase in the CRP level of 10, 50, or 100 mg/L was associated with respective increases of 6, 35, and 82%.

### 3.3. Impact of Inflammation Modulated by CYP2C19-Mediated Metabolism of VRC

We tested the results of the interaction between inflammation and pharmacogenetic markers on VRC Cmin in the three multivariate models (Table 3). Significant interactions were found between CRP and CYP2C19 phenotype in model 2 and between CRP and genetic score in model 3. The evolution of VRC Cmin according to CRP levels stratified according to CYP2C19 genotype is shown for model 2 in Figure 2. The effect of inflammation was reduced for patients with a phenotype of PM/IM, whereas its impact was not significantly different between EM and RM/UM patients.

### 3.4. Quality Assessment

The assessment of study-specific quality scores from the NOS system is summarized in Appendix A. For those studies for which the criterion did not apply, the indication “not applicable” (NA) was entered. The overall quality of the included studies was good (all the studies received a score of 7 or 8 stars).

## 4. Discussion

This meta-analysis performed on a large number of adult patients and VRC Cmin determinations show that the VRC Cmin is influenced by the inflammatory status and genotypes of both CYP2C19 and 3A4, in addition to the age of the patients and the dose of VRC.

The positive and independent association between CRP levels and VRC Cmin is in accordance with the results of numerous previous studies [24,26,27,28] and can be explained by the phenomenon of inflammation-induced phenoconversion [32,33]. Indeed, the expression and activity of CYPs are down-regulated during an acute inflammatory episode, notably under the effects of pro-inflammatory cytokines, such as interleukin-6, which leads to a reduction in CYP-mediated drug metabolism [32,34]. Such an inhibitory effect of inflammation was especially demonstrated in vitro for CYP3A4 and CYP2C19 [32], the main enzymes involved in VRC metabolism in adults [35]. Increases of 10, 50, and 100 mg/L in CRP levels were associated with an increase in the VRC Cmin by 6, 35, and 82%, respectively. For example, an initial VRC Cmin of 1.8 mg/L (median VRC Cmin in this meta-analysis) would increase to 3.3 mg/L for a 100-mg/L increase in the CRP level. This factor is of the same order of magnitude as that found in two European studies [23,36] but larger than that found in a Chinese study that reported an increase in VRC Cmin of 0.6 mg/L [37]. This difference can be explained by different genotypic frequencies between these studies. Indeed, 15.8% of the Asian population is PM for CYP2C19 versus 2.2% of the Caucasian population [38] and only 4.4% in this study.

Concerning pharmacogenetic markers, univariate analysis showed that VRC Cmin tended to be associated with the genotypes of CYP2C19 and 3A4 but was not influenced by that of CYP3A5. Conversely, the combined genetic score, the determination of which integrates all these genotypes, was significantly associated with the VRC Cmin. Multivariate analysis demonstrated a significant impact of the CYP2C19 phenotype on VRC Cmin, without any two-by-two comparisons showing statistical significance, except the RM/UM versus EM comparisons. This result can be explained by the fact that 78% of the patients were presumably Caucasian, and the number of patients with the PM CYP2C19 phenotype was low (only 9/203 patients (4.4%)). Concerning CYP3A4, the trend towards an increased VRC Cmin for IM patients observed in univariate analyses was confirmed in multivariate model 2, in which significance was reached. This finding is in accordance with those of previous studies that demonstrated the impact of genetic polymorphisms of CYP3A4 on VRC exposure [14,19,20,21]. Conversely, we did not find any association between the VRC Cmin and CYP3A5 genotype. This result is consistent with those of two previous studies performed in healthy European volunteers [14,39] but not in agreement with a Chinese study that highlighted a trend towards a higher frequency of the CYP3A5*1/*3 genotype for patients with a low VRC Cmin [21]. These discrepancies may be related to different frequencies of the CYP3A5 genotype depending on the study population. Indeed, the frequency of CYP3A5 expression is relatively low in Europeans, almost 15% (9.9% in this study), whereas 41% of the patients included in the study of He et al. expressed CYP3A5 [21]. Further research is needed to elucidate the impact of the CYP3A5 genotype on VRC exposure.

Two previous studies included in this meta-analysis had suggested that the impact of inflammation on VRC exposure could be modulated by CYP genotypes [27,28]. We obtained a similar result in this meta-analysis, with a smaller effect of inflammation for patients with decreased metabolic capacity for CYP2C19 (IM and PM) in model 2 (significant interaction between inflammation and the CYP2C19 phenotype) than those with normal (EM) or elevated metabolic capacity (RM and UM) (see Figure 2). A study conducted in the Chinese population [37] reported a lower magnitude in the increase in the VRC Cmin in the presence of an inflammatory syndrome than two studies conducted in the European population [23,36]. Such a finding is consistent with our meta-analysis, as a higher frequency of PM for CYP2C19 was found in the Asian population than in the Caucasian population [38].

The fact that VRC exposure appears to be independently influenced by age, inflammatory status, and genetic polymorphisms of both CYP2C19 and 3A4 calls into question the relevance of VRC dose-adjustment strategies based solely on the CYP2C19 genotype [11,12,13]. Although these approaches are useful to reduce the risk of insufficient VRC Cmin in prophylaxis [11,13], their efficiency could be improved by integrating additional determinants [14], particularly the CYP3A4 genotype and inflammatory status. In addition, our findings highlight the fact that interpretation of VRC Cmin measured in routine care and resulting dose adjustment should account for the inflammatory status of the patient [32].

This study is the first to analyze the respective impact of inflammation and pharmacogenetic markers on such a large number of patients and observations. Nonetheless, it had certain limitations. First, among the 46 authors contacted by email, 35 (76%) did not answer, resulting in a still relatively small sample size. In addition, the primary endpoint, namely the VRC Cmin, is an intermediate endpoint, and the consequences of variations of VRC Cmin (due to genetic polymorphisms and inflammatory status) on treatment efficacy and/or adverse effects were not investigated. However, the concentration-effect relationship of VRC is well characterized for both efficacy and toxicity [6], suggesting that any variation in the VRC Cmin would directly influence the treatment outcome. Moreover, we assessed VRC exposure by the VRC Cmin, whereas the ideal parameter would have been the area under the curve of VRC, as recently proposed [14]. Finally, the included studies were heterogeneous in their methodology, with, for example, the absence of CYP3A4/5 genotypes for two of the six studies (representing 67 patients and 266 VRC Cmin determinations), and most of the patients were presumed to be Caucasian, resulting in a small number of PM for CYP2C19.

In conclusion, this meta-analysis demonstrates that VRC exposure is independently influenced by the dose of VRC, age, inflammatory status, and the genotypes of both CYP2C19 and 3A4 aggregated in a combined genetic score. Such findings suggest that an *a priori* VRC dose-adjustment strategy should consider the CYP2C19 and CYP3A4 genotypes, as well as the patient’s inflammatory status. More generally, in the era of predictive, preventive, and personalized medicine, inflammatory markers, already considered to stratify patients with regard to the risk of non-communicable diseases [40,41,42], should be further studied in pharmacokinetics studies. In light of the example of voriconazole, existing strategies of personalized treatment of narrow therapeutic index drugs based most often on one or few pharmacogenomic/demographic parameters could be improved by the integration of additional markers, such as inflammatory markers.

## Figures and Tables

**Figure 1 jcm-10-02089-f001:**
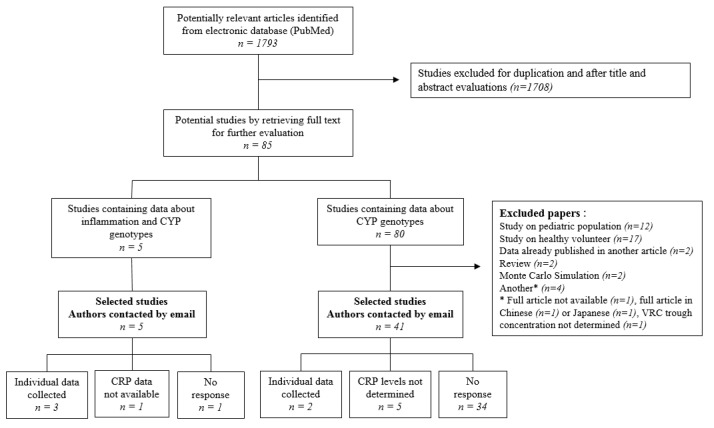
Flow diagram of study selection.

**Figure 2 jcm-10-02089-f002:**
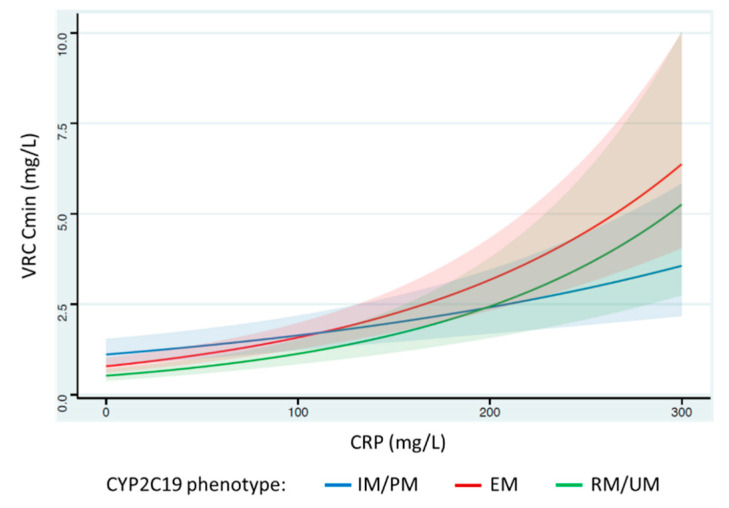
Effects of an increase in CRP levels on the voriconazole (VRC) trough concentration (Cmin) according to the CYP2C19 phenotype. The figure illustrates the predicted increase in VRC Cmin as a function of CRP level from mixed-effects model 2. The blue, green, and red curves represent patients with decreased CYP2C19 metabolic capacity (intermediate (IM) and poor metabolizers (PM)), those with increased metabolic capacity (rapid (RM) and ultrarapid metabolizers (UM)), and those with extensive metabolic capacity (extensive metabolizers (EM)), respectively. Cmin: trough concentration; CRP: C-reactive protein; VRC: voriconazole; CYP: cytochrome P450; EM: extensive metabolizer; IM: intermediate metabolizer; PM: poor metabolizer; RM: rapid metabolizer; UM: ultrarapid metabolizer.

**Table 1 jcm-10-02089-t001:** Studies included in the meta-analysis.

	**Study Design**	**Ethnic Origin, Nationality**	**Number of Patients**	**Number of VRC Cmin Determinations**	**Age (Years)**	**Sex**	**VRC Cmin (mg/L)**	**CRP** **(mg/L)**	**Pharmacogenetic Data**	**Hardy–Weinberg Equilibrium**
Gautier-Veyret et al., 2015 [19]	Retrospective cohort study	Caucasian, French	28	255	52.6 (27.9–61.4)	M: 15 (53.6) F: 13 (46.4)	1.4 (0.1–5.8)	8.0 (3.0–436.0)	CYP2C19 CYP3A4 CYP3A5	Yes
Gautier-Veyret et al., 2019 [25]	Retrospective case-control study	Caucasian, French	57 *	62	60.1 (21.0–79.3)	M: 30 (52.6) F: 27 (47.4)	3.8 (0.1–10.3)	96.0 (3.0–364.0)	CYP2C19 CYP3A4 CYP3A5	Yes
Gautier-Veyret et al., 2020 [26]	Prospective observational study	Caucasian, French	42	150	52.8 (22.3–77.6)	M: 30 (71.4) F: 12 (28.6)	1.3 (0.1–9.7)	43.0 (3.0–369.0)	CYP2C19 CYP3A4 CYP3A5	Yes
Yamada et al., 2015 [30]	Retrospective cohort study	Asian, Japanese	47	47	70.6 (29.4–83.2)	M: 32 (68.1) F: 15 (31.9)	2.4 (0.02–9.6)	1.98 (0.01–26.8)	CYP2C19	Yes
Lamoureux et al., 2015 [15]	Retrospective cohort study	Caucasian, French	9	21	68.0 (21.0–79.0)	M: 7 (77.8) F: 2 (22.2)	0.8 (0.2–4.1)	17.0 (5.0–172.0)	CYP2C19 CYP3A4 CYP3A5	Yes
Veringa et al., 2017 [28]	Prospective observational study	Caucasian, Netherlands	20	219	64.0 (19.0–72.0)	M: 13 (65.0) F: 7 (35.0)	2.8 (0.2–11.2)	59.0 (1.5–401.0)	CYP2C19	Yes
Total	/	/	203	754	58.6 (19.0–83.2)	M: 127 (63.0) F: 76 (37.0)	1.8 (0.02–11.2)	27.0 (0.01–436)	/	/

VRC: voriconazole, Cmin: trough concentration, CRP: C-reactive protein, M: male, F: female. Data are presented as medians (ranges) or numbers (%). * Five patients were included in studies [19,25] with VRC Cmin determined at different times.

**Table 2 jcm-10-02089-t002:** Univariate linear mixed-effect regression analyses.

Covariate	Category	Estimate (95% Confidence Interval)	*p*-Value ^a^
*Demographics* Age Weight *Voriconazole treatment* Daily dose Route of administration *Hepatic function* ASAT ALAT *Inflammation marker* CRP *Comedication* PPI *Pharmacogenomics* CYP2C19 phenotype ^b^ CYP3A4 phenotype CYP3A5 phenotype Combined genetic score ^c^	Per year increase Per kg increase Per mg increase Oral vs. Intravenous Per U/L increase Per U/L increase Per mg/L increase No PPI vs. PPI PM/IM vs. EM RM/UM vs. EM IM vs. EM Non-expressor vs. expressor Per score point increase	0.016 [0.007–0.025] 0.004 [−0.004–0.013] 0.002 [0.001–0.003]−0.318 [−0.497–−0.138] 5.11*10^−5^ [−9.61 × 10^−5^–1.98 × 10^−4^] 3.18*10^−5^ [−1.55 × 10^−4^–2.19 × 10^−4^] 0.005 [0.004–0.006] −0.078 [−0.329–0.173] 0.104 [−0.197–0.406] −0.350 [−0.685–−0.014] 0.374 [−0.058–0.807] 0.309 [−0.155–0.772] −0.558 [−0.853–−0.263]	**<0.001** 0.336 **<0.001** **<0.001** 0.497 0.739 **<0.001** 0.542 0.499 **0.043** 0.093 0.195 **<0.001**

ASAT: aspartate aminotransferase; ALAT: alanine aminotransferase; Cmin: trough concentration, CYP: cytochrome, IV: intravenous, PPI: proton-pump inhibitor, CRP: C-reactive protein, PM: poor metabolizer, IM: intermediate metabolizer, EM: extensive metabolizer, RM: rapid metabolizer, UM: ultrarapid metabolizer, vs.: versus. ^a^ Bold values indicate statistical significance. ^b^ Based on the classification proposed by Moriyama and al. [31]. ^c^ Proposed by Gautier-Veyret and al. [19].

**Table 3 jcm-10-02089-t003:** Multivariate linear mixed-effect regression analyses.

Model (AIC)	Numbers of Observations, Patients, Studies	Estimate [95% Confidence Interval]	*p*-Value ^a^
*Model 1 (AIC = 1805)*	754, 203, 6		
Age		0.014 [0.006–0.022]	**<0.001**
VRC daily dose		0.002 [0.002–0.003]	**<0.001**
CRP level		0.005 [0.004–0.006]	**<0.001**
CYP2C19 phenotype ^b^			**0.016**
PM/IM vs. EM		0.168 [−0.105–0.441]	0.231
RM/UM vs. EM		−0.322 [−0.627–−0.016]	**0.041**
Interaction CRP*CYP2C19 phenotype CRP–PM/IM versus EM CRP–RM/UM versus EM		−9.81*10^−4^ [−0.003–0.001] 1.63*10^−5^ [−0.002–0.002]	0.601 0.340 0.987
*Model 2 (AIC = 1183)*	488, 136, 4		
Age		0.019 [0.009–0.028]	**<0.001**
VRC daily dose		0.003 [0.002–0.004]	**<0.001**
CRP level		0.006 [0.005–0.007]	**<0.001**
CYP2C19 phenotype ^b^			**0.012**
PM/IM vs. EM		0.183 [−0.157–0.523]	0.294
RM/UM vs. EM		−0.366 [−0.691–−0.042]	**0.029**
CYP3A4 phenotype			
IM vs. EM		0.497 [0.128–0.865]	**0.009**
Interaction CRP*CYP2C19 phenotype			**0.019**
CRP–PM/IM versus EM		−0.003 [−0.006–−5.39*10^−4^]	**0.018**
CRP–RM/UM versus EM		7.02*10^−4^ [−0.002–0.004]	0.645
*Model 3 (AIC = 1178)*	488, 136, 4		
Age		0.018 [0.009–0.028]	**<0.001**
VRC daily dose		0.003 [0.002–0.004]	**<0.001**
CRP level		0.006 [0.005–0.007]	**<0.001**
Genetic score ^c^		−0.555 [−0.813–−0.296]	**<0.001**
Interaction CRP*genetic score		0.003 [5.28*10^−4^–0.006]	**0.018**

AIC: Akaike information criterion, VRC: voriconazole, CYP: cytochrome P450, CRP: C-reactive protein, PM: poor metabolizer, IM: intermediate metabolizer, EM: extensive metabolizer, RM: rapid metabolizer, UM: ultrarapid metabolizer. ^a^ Bold values indicate statistical significance. ^b^ Based on the classification proposed by Moriyama and al [31]. ^c^ Proposed by Gautier-Veyret and al [19].

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
