# Peer review of "Combined Impact of Inflammation and Pharmacogenomic Variants on Voriconazole Trough Concentrations: A Meta-Analysis of Individual Data"

_jcm, 2021, doi:10.3390/jcm10102089_

Round 1
Reviewer 1 Report
By performing IPD meta-analysis, the authors demonstrate that voriconazole (VRC) exposure is independently influenced by the dose of VRC, age, inflammatory status, and the genotypes of both CYP2C19 and CYP3A4. The study is well designed, and important findings are revealed.
However, I recommend that the authors address the following points:
Major point
1. The reasons are unclear why authors employed “aspartate aminotransferase” and “alanine aminotransferase” as liver function tests? VRC dose adjustment is usually performed based on Child-Pugh score. The authors should evaluate laboratory values (e.g., total bilirubin) associated with this score.
2. The authors analyzed “weight” and “daily dose” as individual variables. Why did the authors not use the ratio of daily dose (mg)/weight (kg)?
3. Are there any duplicates of patients enrolled in references 19 and 25?
4. Please discuss in more detail how the findings can be translated in clinical practice.
Minor point
1. Spelling out are missing at several instances (for example, VRC in the abstract, ASAT in Table 2). Please review the whole text.
2. Please review the reference format (for example, “J Chemother 2020; :1–11” in reference No. 37).
3. Legend is required for Table S1.
Reviewer 2 Report
The Authors presented a very interesting paper on potential association between inflammation and vericonazole pharmacokinetics, in particular CYP450 family genotypes.
Overall, I have no significant suggestions and /or criticism regarding this manuscript, as - in my opinion - to gather and find out any relationship between polymorphism of CYP450 isoforms and inflammation and the paharmacokinetic parameters for Voriconazole is quite difficult.
My only questions are:
- whether it was possible to catch the information on the time point of the inflammatory state? It might be crucial in order to suggest whether any relathionship exists. In fact the process of inflammation is known to be continously changed within time, as cytokine levels is changed at different time point.
- what type of inflammation was considered ?
- do the Authors have any information whether patients included in the study use any additional pharmacotherapy together with voriconazole? Such data, if any, should be provided in the text
Reviewer 3 Report
In the work, the authors tried to define the impact of inflammation and genetic polymorphisms on exposure to Voriconazole (VRC), an antifungal drug for the treatment of thedangerous invasive aspergillosis, by performing a meta-analysis on individual data. Despite some limitations indicated by the authors themselves (the limited number of studies examined, the heterogenicity of the methodologies, the Caucasian race of the majority of the patients analyzed, resulting in a small number of PM for CYP2C19), the work is well structured. It clearly demonstrated that VRC exposure is influenced by age, inflammatory status (assessed as CRP level) and the genotypes of both CYP2C19 and CYP3A4, the main enzymes involved in VRC metabolism in adults. Therefore, the suggestion to personalize VRC-dose adjustment strategy considering the CYP2C19 and CYP3A4 genotypes, as well as the patient's inflammatory status, is well founded.
Minor points
VRC, CRP should be written in full in the abstract.
Table 1 should be lined up in a more legible manner
Round 2
Reviewer 1 Report
I am satisfied with the modifications made by the authors that have considerably improved the manuscript.
Author Response
We thank the referee for its remark.
Concerning English language and style, our manuscript has been already corrected and edited by an english-native speacker.